# Long-Term Perspectives of a School-Based Intervention to Promote Active School Transportation

**DOI:** 10.3390/ijerph17145006

**Published:** 2020-07-12

**Authors:** Eva Savolainen, Stina Rutberg, Ylva Backman, Anna-Karin Lindqvist

**Affiliations:** 1Department of Health Sciences, Luleå University of Technology, 97187 Luleå, Sweden; stina.rutberg@ltu.se (S.R.); anna-karin.lindqvist@ltu.se (A.-K.L.); 2Department of Arts, Communication and Education, Luleå University of Technology, 97187 Luleå, Sweden; ylva.backman@ltu.se

**Keywords:** active transport, intervention, school-setting, pupils, physical activity, gamification, empowerment, social support

## Abstract

There is a global need for sustainable interventions that increase physical activity among children, and active school transportation (AST) can promote physical activity among schoolchildren. Therefore, an intervention based on gamification, empowerment, and social cognitive theory was initiated in 2016 to promote AST. The aim of this study was to follow up on participants’ experiences one and two years after the AST intervention was initiated. Data were collected through focus groups and individual interviews which were analyzed using qualitative content analysis. Thirty-one pupils (2017), and forty pupils (2018) aged 9–10 years, two teachers (2017, 2018) and one principal (2018) participated in the study. The result is presented as one main theme; “Unity for an active community-An intervention towards making the active choice the easy choice” and three sub-themes; “Well begun is half done-Engagement sparks motivation”, “It takes two to tango-Keep moving with gamifications and togetherness” and “Jumping on the bandwagon–From project to everyday use.” The results show that the concept of the intervention was attractive to re-use and that it created a habit to use AST among the children. Interventions to promote AST can benefit from the use of engagement, togetherness, and gamification.

## 1. Introduction

The World Health Organization [1] underscores the significance of physical activity (PA) for children´s health, but nearly 70% of the world’s children and youths do not reach the recommendation of 60 min of daily PA [2]. In Sweden, only 18% of girls and 43% of boys aged 6 to 11 years accumulate ≥ 60 min of moderate to vigorous PA per day [3,4]. This lack of PA among children is problematic both nationally and globally, and efforts to increase opportunities for PA are very important [2]. Calls for promoting PA have come from global organizations as well as various Swedish authorities [5,6]. According to the curriculum for the Swedish compulsory school [7], the school shall strive to offer daily PA to all pupils. The Swedish National Agency for Education [6] has also suggested that such activities should not only be encouraged during class, but throughout the education program. According to Mitra [8], there are four possible sources of PA among children and young people. One of these is active transport to and from places or activities. Active school transportation (AST) can be a source of regular PA when children and young people go to and from school [8,9,10,11]. AST, in this project, is defined as walking or bicycling whole or part of the way to and from school. The development of sustainable AST interventions is an important issue in both research and politics because it is beneficial from a health, environmental, and transport-safety perspective [12]. 

To promote AST, an intervention built on social cognitive theory was initiated in 2016 at an elementary school class in Northern Sweden. The participants decided to repeat the intervention in 2017 and 2018 and this study presents the follow-up of that. As a result of the intervention, we published five studies including the experience of children, teachers [13], parents [14], winter active school transportation [15], children’s grit [16], as well as described the theory underpinning the intervention [17]. The findings showed that the intervention also created additional value beyond physical activity, such as togetherness, readiness to learn, and changed parental attitudes. It is important to build AST interventions on a solid theoretical framework [18,19], and social cognitive theory is frequently used in interventions to promote PA [20]. Health-promoting processes are quite complex and social cognitive theory is one source for comprehending this complexity [21]. Providing understanding of the interaction between individuals and their environment is the main goal of the theory, and it is suggested that the social environment influences behaviors by making behaviors more or less rewarding in various ways [22]. The Swedish intervention in 2016 was also based on empowerment and gamification [17]. Empowerment is used to promote health and concurrently to respect a person’s right to autonomy, which tends to increase the capacity for independence and coping [23]. Moreover, the teachers integrated assignments from the curriculum into the intervention using gamification [13]. Gamification is described by Deterding, Dixon, Khaled, and Nacke [24] as the using of game design elements in a non-gaming context. Gamification has some common elements similar to social support in behavioral change for health-promotion purposes and has been shown to have high utility in these situations [25]. Likewise, previous interventions have indicated a positive effect of gamification on PA, although further research is needed to fully understand the associations that might enable sustainability [26]; however, a consistent conclusion is necessary for longer follow-ups [19,20,26,27]. Qualitative experiences of AST have received less consideration in the literature concerning AST than quantitative ones [28] even though they are valuable for a better understanding of AST behaviors [29]. Additionally, the value of conducting research with children rather than on children is another important aspect in exploring the complexity of AST behavior [28,29]. It is, furthermore, important to explore the key factors that facilitate an intervention as well as explore the participants’ experience of participating and possible behavior changes due to the intervention. Therefore, the aim of this study was to explore and describe the participants’ experiences one (2017) and two (2018) years after the intervention was initiated.

## 2. Method 

### 2.1. Description of the Intervention 

To promote AST, the intervention started by involving pupils, teachers, and parents in the development of the intervention. For example, children provided the information material to parents, teachers created the gamification elements based on the curriculum, and during a parental meeting, parents came with suggestions about safety issues (children accompanying each other to and from school). Parents also contributed with their ideas and concerns through a survey before the start of the period of using AST. To increase the pupils´ knowledge and inspire them to use AST, workshops regarding health, traffic safety and environment were conducted. During a period of 4 weeks, the pupils used AST. Daily, they measured the class use of AST and they received weekly assignments to solve on their way to school from their teachers. The aim of the weekly assignments was to integrate learning in conjunction with the use of AST. The weekly assignments were based on the curriculum and the knowledge collected by the pupils was used in the lessons. The intervention was introduced in 2016, by the researchers A-K.L., and S.R, and without the involvement of the researchers the teachers and pupils decided to repeat the AST intervention one and two years after the intervention was initiated. The arrangement of the intervention 2017 and 2018 was similar to 2016, except that the inclusion of workshops was left out and different approaches on the weekly assignments were used. 

In 2017, the weekly assignments were to fill a bingo card with different themes. One week, there was a movement of diverse varieties; for example, they had to perform 20 frog hops on their way to or from school in order to mark one square on the bingo card. Another was technology, a third was materials of varied kinds, and a fourth was geometrical shapes in the nature. The pupils also noted how many times they had biked or walked to or from school on a board where they got to mark it with a special stamp that they had picked out. In 2018, the body and senses were in focus. They had two weeks of listening to and seeing objects on their road to or from school. For example, they might have heard a bird singing. During the other two weeks, they were supposed to observe how the surface of different things felt and what feelings they had including how it felt when they biked to school. The assignments of 2018 later evolved into a rap-song and a dance performance with assistance from professionals in dance and singing. The measurements of the daily use of AST 2017 and 2018 were performed by the pupils through making stamps for every AST on a board. In 2016, the pupils placed a sticker on a board for every kilometer they had accomplished.

### 2.2. Study Design and Participants

This work used a qualitative method because it is useful to explore experiences and describe how a social phenomenon is perceived by individuals and groups according to Holloway and Galvin [30]. We used focus groups to learn more about the pupils´ thoughts and feelings about the intervention, which is suitable when a specific topic is explored [30]. Considering the vulnerability of children and the importance of balancing power [31], we tried to facilitate an atmosphere where the pupils could feel free to talk about the intervention, e.g., by telling them that we want to know all about their experience of the intervention no matter good or bad. 

Information about the intervention was sent out to principals of elementary schools in one municipality in northern Sweden [13]. Based on the interest in participating and previous problems with traffic around the school, one school and a class of year one was chosen. In 2016, 42 pupils participated in the intervention. In 2017, twenty girls and eleven boys participated in the focus group discussion. In 2018, nineteen girls and twenty-two boys participated in the focus group discussions. All of the pupils were 9 to 10 years old. The two involved teachers and the principal were also included in this follow-up study. The school ranked among the 10% most socioeconomically privileged nationally, calculated on parents’ educational level, income and the degree of beneficiary. The average distance between home and school was 1.3 km but varied between 0.2 and 6.0 km. All of the pupils and teachers that had extended the AST intervention from 2016 were invited by the researchers during school to discuss and reflect on their experiences. The data-collection was performed during springtime in 2017 and again in 2018. The principal was invited to participate as well in 2018. Parents gave their informed consent during a parental meeting before inviting the children.

This study was conducted in accordance with the ethical principles within the Swedish law for research and The World Medical Association’s Declaration of Helsinki. The study was approved by the Regional Ethical Board in Umeå, Sweden (No.2018-10-31M). Informed consent was given by the participants and the pupils’ parents. Helping the pupils to understand the process and purpose of their participation in focus groups was important in this study. According to Gibson [32], this assistance is also a necessity from an ethical point of view so the pupils can give their consent to participate. In this study, all pupils were informed about their possibilities to participate or not. They were also informed that they could decide to withdraw their participation at any time if they wanted to. 

### 2.3. Data Collection

Data were collected by A-K.L., S.R and Y.B who facilitated the focus group discussion by creating a calm and allowing atmosphere and asking questions as well as asking the participants to reflect about the others opinions to catch as much experience as possible. The focus groups took part during school-hours in the pupils´ class-room. Seven focus groups was performed in 2017 and six focus groups in 2018, with four to seven pupils in each group. The teachers created the focus groups according to the expected likelihood of facilitating an open and dynamic discussion. The two teachers formed one focus group, and the principal was interviewed individually. The focus groups and the individual interview lasted between 16 and 45 min, with an average length of 25 min, and were audio-recorded and then transcribed verbatim. Two different interview guides—one for the pupils and one for the staff—were used to discuss their experiences. The topics in the interview guide for the pupils concerned; (1) what they have done so far, (2) most/least appreciated parts of the intervention and (3) what they have learned. The topics in the interview guide for the staff: (1) what they have done so far, (2) most/least appreciated parts of the intervention and (3) teachable moments. 

### 2.4. Data Analysis

A qualitative content analysis inspired by Graneheim and Lundman [33] was used. From the transcribed interviews, meaning units were selected in accordance with the aim of the study. Thereafter they were coded and compared according to similarities and differences and sorted into preliminary categories. The content in the preliminary categories were compared and discussed, which led to new categories. This procedure was repeated several times, and the final categories were abstracted into themes. These categories were also discussed and renamed a number of times. The original data was recurrently read and compared with the analysis during the process. Notes were collected about the preunderstanding of the authors in order to enhance confirmability as well [34]. The first author was mainly responsible for the analysis, but all authors contributed to all parts of the analysis in order to ensure that varied aspects of the data are visible to enhance trustworthiness. The authors have different professions like physiotherapists, elementary school teachers, and health coaches. They also share an extensive knowledge of qualitative analysis.

## 3. Results

The results were formulated into one main theme and three subthemes (Table 1) and quotes from the focus groups were used to illustrate the conclusions drawn. The quotes are labeled with M for moderator, S1-3 for staff, B1-7 for boys, and G1-7 for girls, followed by the number of the focus group and year. 

### 3.1. Unity for an Active Community-Making the Active Choice the Easy Choice

The main theme “Unity for an active community-making the active choice the easy choice” describes the overall meaning of the participants experiences identified in the subthemes. The sub-themes describe valuable aspects to make an intervention feasible and sustainable. The first subtheme concerns important aspects to get started with the intervention. The second subtheme provides clues for keeping the results from the intervention after its initial phase using the power of groups and collaboration. The third subtheme highlights the influence that the intervention has had on the pupils’ behavior and how they kept the project rolling over time and thus making a school-based AST-intervention sustainable. Hence, by performing the intervention as a community, this work enables AST to become a healthy habit, i.e., the active choice becomes an easy choice.

#### 3.1.1. Well Begun Is Half Done-Engagement Sparks Motivation

In this subtheme, we describe three ways in which engagement, including involvement and active participation in the intervention, promote the participants’ initial motivation according to the staff and the pupils. Opportunities to integrate assignments from the ordinary curriculum and use them to promote learning were considered beneficial to both the staffs’ and the pupils’ motivation. Second, both the staff and pupils emphasized the importance of the pupils’ involvement in decision-making from start and throughout the intervention in order to reinforce the pupils’ motivation; however, this is also a part of the intervention that has room for further improvements since the participants also described a lack of such pupil involvement. Third, both the staff and pupils described that motivation can be increased by others engagement. The three ways in which engagement were perceived by the participants to spark motivation are described in more detail below. 

Being able to create the assignments and choosing a way to measure the active school transports and integrate their ordinary lectures into the intervention was important according to the staff. Their way of integrating the learning process into the intervention evoked both the pupils’ and the staffs’ engagement to repeat the intervention in both year two and three. The pupils stated that it was really fun and that they had learned a lot. Likewise, the staff described that instead of feeling overwhelmed by performing the intervention, it had created interest in how the concept could be used in other subjects. 

S1:
*It has become a natural part of the school day: We have integrated it into everyday teaching.*


S2:
*Yes*


S1:
*It is not something I apply on top of it all it is included with the other subjects so that it will be neither a burden to the pupils nor to us; it does not feel made up.*


M:
*How has it been integrated into the teaching?*


S1:
*Well, if we work with natural science or social science, then we will set up assignments from these subjects into the project. (Staff, 2018)*


The staff described the pupils’ involvement in how the intervention was executed could be improved to enhance their engagement, but they expressed uncertainty about how it would be performed. Furthermore, they believed that the assignment structure should be decided by teachers since they have knowledge of what curricular demands there are on the pupils. However, they expressed an advantage of making the pupils feel involved and owning the project. Likewise, the pupils stated that it felt better when they were more involved in the assignments and got to decide different things, e.g., the assignment in 2018, where they created a song and dance together with the staff. Some children said that the song and dance were created jointly, while others did not feel that they had been involved enough. At the same time, they did not know what their weekly assignments were leading to in the beginning when they collected words, feelings, and movements experienced during their AST. Not knowing this from the beginning was an advantage for some. It was thought to have brought pressure to perform something great while others expressed that they would have preferred to know its purpose earlier.

M:
*Is it more fun when you yourself have contributed to the assignments?*


Several:
*Yes*


B3:
*So they don’t say you should be that and you should be this*


G2:
*Yes, because then WE have done a dance then it is not just them; everyone has contributed.*


B1:
*And the song too (Focus Group 4, 2018)*


The pupils also highlighted the importance of the engagement of the staff; they felt that the staff’s dedication and enthusiasm contributed to their commitment in doing assignments and using AST. In addition, the staff thought that it was important to feel dedicated in order to influence the pupils and implement the intervention. However, the intervention was described as more of a light version because of a lack of time due to national tests that was planned and executed during the last year: this lack of time affected their engagement in the intervention and was also noticed by the pupils. The staff also stated that not all teachers could be interested in doing this kind of intervention, but then the pupils can inspire each other instead. On the other hand, the pupils reflected about other pupils’ engagement and felt that others’ lack of motivation affected their joy and motivation to perform their best in the assignments.

S1:
*We [the teachers] have a very important role*


S2:
*Yes*


S1:
*To be enthusiastic and inspirational because it rubs off on them*


S2:
*Absolutely, yes*


S1:
*But if you think why should I do this and*


S2:
*Yes, if a teacher feels that way, then maybe the children can inspire each other instead*


M:
*There must be someone that can do it*


S2:
*Yes, there must be, but if there is none, because we are different and some really think it is a plague; you might be lucky, and the children can bring out the enthusiasm instead*
*(Staff, 2018)*


Having a supportive organization where teachers support each other and having support from the management to spend time on doing the project were also important for the engagement according to the staff. This was especially important in the beginning of the intervention, where it takes extra time to plan things out of the ordinary. In addition, they suggested that more staff at the school could be engaged in the intervention including physical education teachers, school nurses, and leisure-time centers. Furthermore, the pupils as well as the staff talked about the engagement of people outside ordinary schools in the project including policemen, researchers, and experts on the environment that created memorable moments in workshops during year one. The staff also expressed a possibility of engaging parents in the intervention because they too could be an inspiration for the pupils.

#### 3.1.2. It Takes Two to Tango-Keep Moving with Togetherness and Gamification

This sub-theme identifies three main strategies for maintaining a successful intervention after its initial phase as perceived by the participants. The first regards the importance of allowing for prosocial behavior during the assignments and the promotion of togetherness. The second concerns the importance of having different AST assignments to address differences in pupils’ interests, competence levels, and ways of learning. The third regards the relevance of measuring and making visible group progress as well as providing rewards after fulfilling whole-class goals. Below, these three perceived main strategies are described.

Both pupils and staffs stated that one important aspect of the intervention was that it was carried out with the whole class, which enabled them to share the experience and success with each other. Having the possibility to contribute to the success of the whole class, was something that was described by the staff as an important aspect—especially for pupils that might have difficulties in other areas. Moreover, the staff stated that it probably was a good idea to pair the pupils when they did it the first time. They then made contact with each other in an early stage and could later choose to keep walking or biking with the same pupil, which some of them did. The importance of the social aspects was further underlined by the pupils in their talk about the value of having a friend to walk or bike with. They considered it more fun to bike or walk together, which also initiated an opening to talk about private things, share some laughs, and even compete on their way to school. 

B2:
*I think it feels like it goes faster [when you have a friend with you], because you don’t think of the time*


G2:
*You don´t think that it takes a long time*


B2:
*Yes*


G2:
*You think of talking to your friend instead (Focus Group 5, 2017)*


Another reason for walking or biking with a friend was that a friend could be helpful if an incident would occur on the road. A friend could also help with the assignments. However, sometimes it was satisfying to bike or walk alone and not being forced to adjust to another’s speed according to the pupils. The staff also reflected upon the social effects of the intervention:

S2:
*I´m thinking of the class, I think this contributes to a more welded class, a sense of “we.” They feel happy and alert, they have something in common. I don´t really know how to express myself but there are so many things in this that can transfer something more into the otherwise daily work. That the profits have to be seen from a longer perspective: you do it for the future I think (Staff, 2018)*


According to the pupils, there was something in the assignments for everyone to remember as a favorite. Some pupils expressed that the group assignments during the first year of the intervention were the most fun while others favored the bingo-cards with individual tasks during the second year. The creation of the song and dance was considered to be the most exciting assignment by some pupils. Commonly, the pupils expressed that the assignments need to be sufficiently challenging—not too difficult, but not too simple. Looking for different things on their way to school during the third year of the intervention was considered to be too easy by some. This led to a lack of commitment while others found that the bingo-card during the second year in some sense had been too hard to accomplish; thus, they did not finish it. Nevertheless, the pupils appreciated getting assignments in the form of practical tasks as they found it more enjoyable in comparison with reading a book or watching a screen. In addition, the teachers talked about this project being an opportunity to practice teaching in several different ways giving many possibilities to learn. 

S1:
*I think it has been good to get input from different ways, like auditory and visually and also have study visits and doing things with the body. I think it positively affected their learning, and they have something to remember I think*


M:
*Well it sounds like the things you pick out are the kind of things that stand out from everyday life, so is that what you think?*


S1:
*Exactly. It strengthens their learning besides having just read it in a book.*


S2:
*And that is what we should use in our teaching, arrange different types of inputs as much as possible since everyone learns in different ways (Staff, 2018)*


Both the pupils and the staff described an eagerness among the pupils to put a stamp on the board and to do the assignments. The pupils also expressed that it was more fun when they received attention for their effort to use AST like being able to put a mark on a board that was visible to all in the class. Additionally, some thought that it was thrilling to see if they all could use AST in one day during the intervention and accomplish a full board that day. They also reflected back on the first year of the intervention when they had measured the number of kilometers that was marked on a map to show how far they had travelled together, and they stated that this had encouraged them to use AST more. Moreover, many stated that the attention they received in year one (when they received a joint badge on the board for accomplishing a weekly assignment) was more motivating than the individual assignment the second year of the intervention with a bingo card without a reward. On the contrary, the teachers had some doubts after the second year towards giving rewards considering that everyone might not have the possibility to participate all the time. This had them wondering if they could have added a joint reward to the class even though it was an individual assignment. The pupils further stated that it was exhilarating to partake in a celebration when they had finished weeks of assignments. Some of them described this as the best part of the intervention, although they did not consider it to be the most important part of it.

B1:
*It is exciting when you come to school and see how many there are that have got a stamp*


G2:
*And it is fun to get a stamp and show the class that you have walked or cycled (Focus Group 1, 2018)*


#### 3.1.3. Jumping on the Bandwagon—From Project to Everyday Use

In this sub-theme, we provide a description of the participants’ reflections about the effects of the intervention, and how this motivates them to keep going with the new mode of transport and new habits over time. This is three-fold and is in regards to increased habituated PA among the pupils, the schools and the surroundings’ adaptation towards AST, and less stress and higher concentration during class. Herein, these three parts are described.

Most of the pupils stated that they after these three years of participating in the intervention walked or biked all year with minor exceptions and that it had become a habit to use AST and that they did not reflect much on their choice of transport anymore. Even if some were driven by car on occasions or took the bus to and from the school, they could get off a little further away from school and the added active route could then be counted as an AST according to the pupils. In addition, they retrospectively reflected on the effects of the intervention and concluded that they probably did bike or walk more than they would have done without it. Concurrently, some pupils had no choice but to do it, as their parents refused to drive them and some of the pupils said that they would simply not ride a car because they had learned that it was bad for animals and nature. The pupils also announced that they will continue to use AST in the future.

B1:
*It’s like a habit for me now to cycle and walk*


B2:
*A habit for me to come home by myself*


G2:
*I live so close but those who may live [far from school] on the X*


G1:
*I also live there but I don’t think it´s far, I just do it [bikes]*



*(Focus Group 2, 2018)*


The staff stated that the intervention had led to less traffic around the school, especially when more classes at the school became involved. This was an appreciated effect since the traffic situation around the school was a strong reason for the school to partake in the intervention from the beginning. The staff identified some improvements to make it easier to implement the intervention such as making space for parking the bicycles especially during wintertime. The staff also talked about their responsibility to maintain moral values in accordance with the curriculum and their perception of the intervention as a plausible way of contributing to it. Moreover, the staff declared that the intervention had become a part of the school’s concept and is in fact attracting parents to choose their school nowadays. 

S3:
*I see that there are many more parents, well this is not a scientifically conclusion, but my reflection of it is that I see a lot more parents cycling now. And before we have had a lot of problems with parking in the morning, but now we have no reactions concerning the traffic in the morning; no one is saying anything, and we have empty parking spaces too. (Staff, 2018)*


Having been included in the intervention was something that the pupils were proud of, which also led to a will to pass it on. Hence, the pupils described themselves acting like ambassadors for AST. The pupils also described several perceived effects of AST, e.g., they felt liberated being able to bike or walk to and from school and they felt less stressed when they did not have to ride with their parents to school and could stay longer at home in the morning. AST also gave the pupils a sense of general well-being, and doing the assignments on their way to school was described by both the staff and the pupils as a good approach to prepare for a day in school. In addition, both the pupils and the teachers had noticed that they (the pupils) were more alert and could concentrate better during school. If they had been talking to a friend on their way to school, then they had less need to chit-chat during lessons. 

G3:
*and when you have a long shift and have to count math, you get better concentration when walking or cycling (Focus group 6, 2017)*


## 4. Discussion

The result provides a longer follow up and adds knowledge about the participants’ experiences of taking part in the intervention, which is important because there is a lack of such research [26]. This is also useful because it is important to identify the determinants for AST to promote sustainable AST behavior [35]. Our result shows that the children had developed a habit of using AST and that the active choice had become an easy choice and was not something that the pupils reflected much upon anymore. This is in line with Karppinen et al. [36], who concluded that PA interventions should create a habituated behavior in order to maintain a change. This intervention was school-based and formed a unity of school and family, which is fundamental for a successful AST intervention according to Burgueño et al. [35]. In addition, previous studies regarding this intervention have shown that parents, who are the key-holders to their children using AST, became more positive to AST after being involved in the intervention [14]. The results also show that the intervention was attractive for the school to re-use year after year and it was easy to integrate assignments from the curriculum. This is important because it has been concluded previously that this integration can contribute to lower rates of exhaustion [20], and that school-based interventions need to be feasible and not consume too much time [37]. 

The results show that engagement, togetherness and gamification were important aspects to create motivation to a behavioral change in this project. Motivation is vital to succeed with a behavioral change regarding AST and promoting extrinsic motivation in the beginning of an intervention has shown to be positively associated with increased AST behavior [35]. The self-determination theory (SDT) explains how, e.g., different forms of extrinsic motivation could turn into autonomous motivation in order to facilitate and maintain a behavior change [38]. We found that the feeling of engagement was expressed as one central part in the beginning of the AST intervention. This is an important result because active engagement enables feasibility and enhances the likelihood of a successful and sustainable intervention [39]. The staff’s dedication and enthusiasm contributed to the pupils’ commitment in doing assignments and using AST. Engagement of the teachers was generated by the opportunities to integrate assignments from the ordinary curriculum and use them to promote learning, which were considered beneficial to both the teachers’ and the pupils’ motivation. The result also showed that one way of creating engagement was to involve participants in the decision-making process concerning the intervention, which is in line with an empowerment-based perspective [40]. Moreover, in accordance with SDT, to facilitate motivation, it is important for people to synthesize the meaning of the intervention with respect to their own goals and values—this process is promoted by a sense of autonomy and allows individuals to actively transform values into their own [38]. This is also in line with the description of the effects of empowerment by Alsop and Heinsohn [41] and core determinants in SCT (Social cognitive theory), e.g., outcome expectations and goals [21]. Although our results indicate that the staff were highly involved in decision-making, the results also point out that there is room for improvement regarding the pupils’ involvement to further increase motivation. 

The result shows that togetherness was another essential part of the intervention which could be viewed from the perspective of SCT, i.e., that a behavior is partly affected by the social response to the behavior [21]. This is also in line with Harris [26], who found that togetherness was a motivating component to both commitment and successful behavior change. A review pinpoints the importance of social support and modelling in PA interventions [20]. In addition, a previous study found that social aspects were the most important for children regarding different transport modes [42]. Working with group assignments and sharing the effort and success were motivating according to the participants. This could be explained by SDT where the sense of relatedness is important when it comes to facilitating extrinsic forms of motivation growing into autonomous motivation [38]. According to Ryan and Deci [38], the primary reason for people to perform extrinsically motivated behaviors that are not in their interest is because the behavior is prompted, valued, or modeled by significant others. The authors further discuss how the social environment influences the processes of non-intrinsic motivated behavior that can become self-determined in accordance with the basal psychological needs for motivation [38]. For instance, Burgueño et al. [35] concluded that extrinsic forms of motivation such as integrated regulation and identified regulation in SDT are positively associated with AST behavior and that this is important to consider when designing interventions to promote AST. Furthermore, having different AST assignments that address the differences in pupils’ interests, competence levels, and ways of learning affected the participants to keep doing them. This is in line with one of the key structures of gamification: The capacity to overcome challenges by growth, learning, and development [25]. Similarly, in SDT, the function of perceived competence is important for motivation [38]. Furthermore, a previous study concluded that gamification is useful to address motivation in a school context [43]. Additionally, the criteria for SDT (competence, relatedness and autonomy) can be used to judge the usefulness of gamification to address motivation [43]. 

Overall, these results, i.e., the engagement, togetherness and gamification, were important aspects to create motivation, underscore the importance of motivational considerations when explaining AST behavior among variables for AST such as parental worries. The perceived intervention effect on the traffic situation around the school with fewer cars and a safer environment is an important finding since a safe environment is a prerequisite for AST in order to cope with parental worries [44]. The results show that the intervention actually increased the attractiveness of the school and made more parents interested in placing their children there. An additional valued effect of using AST was the participants’ experience of higher concentration and more alertness during class; similar effects have been described previously [45]. In addition, our result show that the participants experienced less stress and a sense of wellbeing which is in line with health benefits of PA and AST that have been shown before [8,46,47].

### Strengths and Limitations

We chose a qualitative approach, focus groups, and content analysis to explore and describe the participants’ experiences one (2017) and two (2018) years after the AST intervention was implemented. A qualitative design was chosen because it gives the participants an opportunity to speak freely about their perception and feelings towards the intervention [48]. Furthermore, the imbalance between young people and adults are managed with focus groups by transferring the balance of power toward participants and away from the researcher [49]. In this study, the teachers tried to create groups according to the likelihood of facilitating an open and dynamic discussion since they were familiar with the pupils. However, there is a probability that individual interviews could have resulted in other data [48]. Although we cannot be certain that individual interviews would not give other views of the intervention, we feel confident that the material provides versatile aspects of the participants experiences. Furthermore, all authors played an active role in the analysis of the data to improve confidence in the accuracy of the data and the interpretations and thereby increase trustworthiness [33]. Consequently, the result was viewed from various angles by a research team with different professional backgrounds. This result is supported with quotations to further enhance the credibility [33]. An interview guide was used so that the questions were in the same areas and contribute to consistency during the data collection. This approach could reinforce dependability, according to Graneheim and Lundman [33]. Although we may consider these results to be highly valuable, one must consider the fact that the effects are dependent on the context and the users [50]. The context and the participants of this study are described with caution in order to maintain the participants’ confidentiality. The analysis is described in detail to enhance transferability. Our presentation of the findings and appropriate quotations is another way of enhancing transferability [33]. Nevertheless, we leave it to the reader to judge the transferability to another context. 

A possible limitation of this study is that the pupils’ statements, like getting better concentration of AST, could be a consequence of being taught about possible effects of physical activity and not their own experience. To minimize this, the researcher that facilitated the focus groups asked the participants to give examples of what they meant with their statements. It is also a limitation in this study that the included school was ranked among the 10% most socioeconomically privileged nationally and more studies are needed in other contexts. There is also a further need for research within this field as physical activity and AST decreases along with an increasing age of children. Longitudinal studies of children’s adherence to physical activity as a result of the application of the intervention would therefore be valuable. Moreover, there is a need to strengthen the strategies to get more children to use AST and more knowledge about parents’ choices to let their children use or not use AST.

## 5. Conclusions

The results show that the concept of the intervention was attractive to re-use, both from the perspective of staff and pupils, the year after and two years after the first time. Moreover, the result shows that the participants developed a habit of using AST and that the intervention could therefore be a source of increased physical activity. Moreover, the traffic around the school decreased, which could be a valuable incentive for schools to engage in the intervention. Interventions to promote AST can benefit from the use of engagement, togetherness, and gamification. 

## Figures and Tables

**Table 1 ijerph-17-05006-t001:** Overview of the results.

Main Theme	Subthemes
Unity for an active community-Making the active choice the easy choice	Well begun is half done-Engagement sparks motivation
	It takes two to tango-Keep moving with togetherness and gamification
	Jumping on the bandwagon-From project to everyday use

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
