# Peer review of "Long-Term Perspectives of a School-Based Intervention to Promote Active School Transportation"

_ijerph, 2020, doi:10.3390/ijerph17145006_

Round 1
Reviewer 1 Report
Abstract
The abstract adequately summarizes the research by outlining the objective, the methodology, the procedure of data collection, and the results provide the main findings. However, the number of subjects interviewed and the average age or range of the total sample are not specified.
Keywords:
The selected keywords are suitable and will allow a quick search by interested researchers. I would add "schoolchildren or pupils" to the list to facilitate researchers' searches.
Introduction
The general idea of the research is presented with an adequate and updated theoretical framework. But it should be defined concisely and precisely as "Active School Transportation (AST)". Although it is a familiar term in some countries, in others it is not. So, what do you mean by that?
The referenced research is from the last few years.
Materials and Methods
The sample was adequate and focused on students, teachers and parents as the first ones involved in "Active School Transportation (AST)".
Data analysis
They deepen the qualitative content analysis with the appropriate use of the focus group.
Results
The results are sufficiently developed. By raising the positive and negative aspects to each issue.
Discussion
Well presented by the authors.
However, the statement reflected in lines 445 and 446: "we found that the participants themselves experienced health benefits like less stress and a general sense of well-being" lacks any scientific evidence to justify it. Therefore, it is recommended that something similar to the following example be used: "as expressed by the participants in this study, it could be said that they feel less stressed and have a better sense of well-being when they use the "Active School Transportation (AST)" program.
Conclusions
They are clear and concise.
Strengths and Limitations
It makes clear the need to go deeper into some aspects not covered.
Future Lines of Research
I would add this section within the overall structure of the paper.
Including approaches such as:
- Study and analysis of children's adherence to physical activity as a result of the application of the "Active School Transportation (AST)" program through a longitudinal study (5-10 years). Perhaps one of the most complicated times is the adolescence of these children.
- To propose the future development of strategies for the application of "Active School Transportation (AST)" in children who go to and from school by bus, car, etc. Since there are children who for various reasons will not be able to walk or cycle to school.
In general
The article is interesting because it offers a proposal for real application as a measure to create healthy habits for children and even their families. In this sense, the most interesting aspect is the creation of adherence to these routines throughout the life of the participants.
It responds to the aim and scope requested in the International Journal of Environmental Research and Public Health (IJERPH): Children's Health, Health Behavior, Chronic Disease and Health Promotion or Exercise and Health among others.
Author Response
We appreciate the opportunity to improve our manuscript and we are grateful for your time and effort to make all of these remarks helping us to do that. Your comments are in bold and our answers are in italic.
Sincerely,
Eva Savolainen, Stina Rutberg, Ylva Backman and Anna-Karin Lindqvist.
Abstract
The abstract adequately summarizes the research by outlining the objective, the methodology, the procedure of data collection, and the results provide the main findings. However, the number of subjects interviewed and the average age or range of the total sample are not specified.
We have updated the information both in abstract and under 2.2 study design and participants
Keywords:
The selected keywords are suitable and will allow a quick search by interested researchers. I would add "schoolchildren or pupils" to the list to facilitate researchers' searches.
We have added pupils to the keywords
Introduction
The general idea of the research is presented with an adequate and updated theoretical framework. But it should be defined concisely and precisely as "Active School Transportation (AST)". Although it is a familiar term in some countries, in others it is not. So, what do you mean by that?
Thank you for this comment. We have added a sentence in the introduction about our definition of AST in this project.
The referenced research is from the last few years.
Materials and Methods
The sample was adequate and focused on students, teachers and parents as the first ones involved in "Active School Transportation (AST)".
Data analysis
They deepen the qualitative content analysis with the appropriate use of the focus group.
Results
The results are sufficiently developed. By raising the positive and negative aspects to each issue.
Discussion
Well presented by the authors.
However, the statement reflected in lines 445 and 446: "we found that the participants themselves experienced health benefits like less stress and a general sense of well-being" lacks any scientific evidence to justify it. Therefore, it is recommended that something similar to the following example be used: "as expressed by the participants in this study, it could be said that they feel less stressed and have a better sense of well-being when they use the "Active School Transportation (AST)" program.
Thank you for this remark, we have rewritten the sentence with inspiration of your suggestion.
Conclusions
They are clear and concise.
Strengths and Limitations
It makes clear the need to go deeper into some aspects not covered.
Future Lines of Research
I would add this section within the overall structure of the paper.
Including approaches such as:
- Study and analysis of children's adherence to physical activity as a result of the application of the "Active School Transportation (AST)" program through a longitudinal study (5-10 years). Perhaps one of the most complicated times is the adolescence of these children.
- To propose the future development of strategies for the application of "Active School Transportation (AST)" in children who go to and from school by bus, car, etc. Since there are children who for various reasons will not be able to walk or cycle to school.
We appreciate these remarks and have added text about future research in line with your suggestion in the section.
In general
The article is interesting because it offers a proposal for real application as a measure to create healthy habits for children and even their families. In this sense, the most interesting aspect is the creation of adherence to these routines throughout the life of the participants.
It responds to the aim and scope requested in the International Journal of Environmental Research and Public Health (IJERPH): Children's Health, Health Behavior, Chronic Disease and Health Promotion or Exercise and Health among others.
Reviewer 2 Report
This is a report of qualitative data collected from pupils and teachers involved in an active school travel intervention. Data were collected by focus groups (children) and one interview and focused on experiences of the intervention. Some interesting insights on experiences of the intervention were gathered. However, I have concerns about the description of the intervention, the description of the methods, the quality of the data and the way in which the data were interpreted. The discussion is difficult to follow in places.
I am concerned that the first part of the title over-claims based on the findings of this study. It is not clear that this intervention was responsible for "making the active choice the easy choice", as is implied here. It would be useful to include a basic description of the type of intervention in the title. I am not convinced that the term "long-term perspectives" in the title is justified (see below).
In the introduction I would have liked to have seen more detail of the intervention, and of previous publications on the intervention, which are cited in the paper. Are there quantitative data showing whether or not the intervention had an effect on active school travel? Or was this not assessed? Also, a stronger justification of the need for qualitative evidence on experiences of the intervention is needed.
The suggestion on lines 66-67 that the aim of this study was to explore and describe the participant's experiences one and two years after the intervention was implemented is confusing. The authors note that the school repeated the intervention during those years, and much of the data concerns how the intervention was delivered in those subsequent years.
The description of the intervention offered in the methods section is very hard to understand. I would like to know exactly what the intervention involved, and how it changed over the years. In addition more detail is needed. In particular (but not only), what was the aim of the 'assignments' and how did they work? How was the intervention integrated with the curriculum (something that is emphasised throughout, but not explained)? How did the intervention change over the years? Who delivered the intervention in its first year? Were the authors of this study involved then? What were parents "informed about" (line 76)? How was AST measured? What was the effect of the intervention?
lines 81-82 - Why are the focus groups mentioned here, in the section describing the intervention?
More justification for the use of focus groups is needed. There are other ways of ethically collecting data from children (lines 100-101). The limitations of focus group data should be considered in the paper.
Were the 40 children who were participants in this study involved in the intervention in the same year that data was collected?
13 focus groups are mentioned (Line 121), with 4-7 children in each. This suggests the involvement of more than 40 children. How many teachers participated in the study? What was the average duration of the focus groups? Who collected the data? How were the focus groups facilitated?
I found the topic guides troubling in that the children and teachers were asked about "the best parts of the intervention" (lines 127-128) but not about the worst, or parts they did not like. How wide-ranging was the discussion based on these prompts? How did the researchers prompt conversation between participants on the intervention?
What is meant by a "meaning unit" (line 134)? This is probably a matter of finding a better translation into English. What does "meaning units were selected in accordance with the aim of the study" mean? (line 134).
What is meant by "crossover effects from high engagement to higher motivation" (line 175)?
It was not always obvious to me that the quotes chosen supported the statements made by the authors. For example the meaning of lines 229-244 is rather unclear.
The "social aspects of the intervention" were emphasised (lines 260-261 in particular). Was the intervention inherently social? This needs to be made clear in the description of the intervention in the methods section.
What was exciting for children about "measuring the amount of AST"?
You state that most of the pupils said they walked or biked all year (line 331). This is important background to the study and should be stated earlier, ideally with quantitative data to support the statement. Are you suggesting that this was to some extent a consequence of the intervention?
The authors should be careful to state the views of the study participants rather than accept them as fact. For example, on line 338-9 they suggest that "some of the pupils would simply not ride a car because they had learned that it was bad for animals and nature". This should be reported as "some of the pupils said that they would simply not ride a car...".
The authors should think about whether children (and perhaps teachers) sometimes repeated discourses that they had been taught as part of the intervention (e.g. "you get better concentration when walking or cycling", lines 374-375.). This should be considered in the interpretation of the data, and mentioned as a limitation in the discussion, particularly if the same researchers were involved in originally delivering the intervention and in collecting these data (but even if they were not).
On line 367 children are reported to have "felt less stressed when they could stay longer at home in the morning". Can you explain why they would have been able to stay at home longer?
The authors should ensure that the discussion is closely tied to the results. For example, I am not convinced the the results reported here show that "the children had developed a habit of using AST" (line 380) as a consequence of the intervention, which is implied.
Line 437 states that "Overall, these results underscore the importance of motivational considerations when explaining AST behavior". Can you explain more clearly what this means, and how you reach this conclusion?
The study limitations section should consider the implications of the fact that the school was attended by children of high SES.
Author Response
We appreciate the opportunity to improve our manuscript and we are grateful for your time and effort to make all of these remarks helping us to do that. Your comments are in bold and our answers are in italic.
Sincerely,
Eva Savolainen, Stina Rutberg, Ylva Backman and Anna-Karin Lindqvist.
This is a report of qualitative data collected from pupils and teachers involved in an active school travel intervention. Data were collected by focus groups (children) and one interview and focused on experiences of the intervention. Some interesting insights on experiences of the intervention were gathered. However, I have concerns about the description of the intervention, the description of the methods, the quality of the data and the way in which the data were interpreted. The discussion is difficult to follow in places.
I am concerned that the first part of the title over-claims based on the findings of this study. It is not clear that this intervention was responsible for "making the active choice the easy choice", as is implied here. It would be useful to include a basic description of the type of intervention in the title. I am not convinced that the term "long-term perspectives" in the title is justified (see below).
Thank you for your thoughts about this, we have removed “making the active choice the easy choice” and added that it is school-based, however we think that following this class for 3 years is a long-term follow-up and therefore we would like to keep it.
In the introduction I would have liked to have seen more detail of the intervention, and of previous publications on the intervention, which are cited in the paper. Are there quantitative data showing whether or not the intervention had an effect on active school travel? Or was this not assessed? Also, a stronger justification of the need for qualitative evidence on experiences of the intervention is needed.
We have clarified some details about the interventions and added text about the result of the previous publications on the intervention in the introduction. In this study we did not collect any quantitative data showing the effect on active school travel. We are planning for a RCT study and this study will provide important knowledge that we take into account for that study. We have added text in the introduction for a stronger justification of qualitative evidence.
The suggestion on lines 66-67 that the aim of this study was to explore and describe the participant's experiences one and two years after the intervention was implemented is confusing. The authors note that the school repeated the intervention during those years, and much of the data concerns how the intervention was delivered in those subsequent years.
We appreciate this comment and have acknowledge that this could be confusing. We have rewritten parts in the last section of the introduction to make this clearer.
The description of the intervention offered in the methods section is very hard to understand. I would like to know exactly what the intervention involved, and how it changed over the years. In addition more detail is needed. In particular (but not only), what was the aim of the 'assignments' and how did they work? How was the intervention integrated with the curriculum (something that is emphasised throughout, but not explained)? How did the intervention change over the years? Who delivered the intervention in its first year? Were the authors of this study involved then? What were parents "informed about" (line 76)? How was AST measured? What was the effect of the intervention?
Thank you for acknowledging this aspect of uncertainty. We have rewritten the whole section to clarify how the intervention was performed and how AST was measured..
lines 81-82 - Why are the focus groups mentioned here, in the section describing the intervention?
Thank you for making us aware of this, we have removed that sentence since it belongs in the section; data collection, were it is also mentioned.
More justification for the use of focus groups is needed. There are other ways of ethically collecting data from children (lines 100-101). The limitations of focus group data should be considered in the paper.
We have added text in the section; data collection in accordance with your remarks. The limitations of focus groups are considered in the section; strengths and limitations.
Were the 40 children who were participants in this study involved in the intervention in the same year that data was collected?
Yes, the participants in this study were involved in the intervention the same year that data was collected and hopefully the rewritten part of the description of the intervention makes this clear.
13 focus groups are mentioned (Line 121), with 4-7 children in each. This suggests the involvement of more than 40 children. How many teachers participated in the study? What was the average duration of the focus groups? Who collected the data? How were the focus groups facilitated?
We have clarified how many pupils that participated each year. As the same children participated both in 2017 and 2018 the total number of children is not more than 40. There were two teachers that participated in the study and we have described this under the section: study design and participants. The average duration was 25 minutes and we have added text in the data collection part to clarify this. We have also clarified who collected the data and how the focus groups were facilitated in the data collection section.
I found the topic guides troubling in that the children and teachers were asked about "the best parts of the intervention" (lines 127-128) but not about the worst, or parts they did not like. How wide-ranging was the discussion based on these prompts? How did the researchers prompt conversation between participants on the intervention?
We appreciate that you found this error. We asked about both positive and negative parts of the intervention. We have added text in this section; data collection to show that we asked about them both.
What is meant by a "meaning unit" (line 134)? This is probably a matter of finding a better translation into English. What does "meaning units were selected in accordance with the aim of the study" mean? (line 134).
The reference, Graneheim and Lundman uses the concept “meaning unit” to describe a sentence or a small section of text that visualize an aspect of the result. These units is collected from the transcribed text to answer the aim of the study. We would like to keep the word “meaning unit” in accordance to the reference.
What is meant by "crossover effects from high engagement to higher motivation" (line 175)?
This has been re-written to clarify what it means.
It was not always obvious to me that the quotes chosen supported the statements made by the authors. For example the meaning of lines 229-244 is rather unclear.
Thank you for noticing this, the quote that you specified appear to be in the wrong place and has been moved one paragraph up to make the support of statements more obvious.
The "social aspects of the intervention" were emphasised (lines 260-261 in particular). Was the intervention inherently social? This needs to be made clear in the description of the intervention in the methods section.
Thank you for making us aware of this, we have adjusted the sentence as we realized that the wording was not in accordance with the meaning of the sentence.
What was exciting for children about "measuring the amount of AST"?
We have changed the sentence to clarify this.
You state that most of the pupils said they walked or biked all year (line 331). This is important background to the study and should be stated earlier, ideally with quantitative data to support the statement. Are you suggesting that this was to some extent a consequence of the intervention?
This is stated by the participants as a consequence of the intervention and we have in the result section added text to clarify this. We have not collected any quantitative data in this study so unfortunately we can not support these statements with that.
The authors should be careful to state the views of the study participants rather than accept them as fact. For example, on line 338-9 they suggest that "some of the pupils would simply not ride a car because they had learned that it was bad for animals and nature". This should be reported as "some of the pupils said that they would simply not ride a car...".
This has been changed in accordance with your suggestion.
The authors should think about whether children (and perhaps teachers) sometimes repeated discourses that they had been taught as part of the intervention (e.g. "you get better concentration when walking or cycling", lines 374-375.). This should be considered in the interpretation of the data, and mentioned as a limitation in the discussion, particularly if the same researchers were involved in originally delivering the intervention and in collecting these data (but even if they were not).
We appreciate this comment as it made us reflect about the result again and we have clarified the limitation this aspect gives to the result under the section strength and limitations. Rereading the interviews we acknowledge that we have tried to ask clarifying questions and asked for examples to explain the statements to minimize statements that is not their own experiences.
On line 367 children are reported to have "felt less stressed when they could stay longer at home in the morning". Can you explain why they would have been able to stay at home longer?
Because they did not ride to school with their parents that drove to work earlier in the morning. We have added this information in the text.
The authors should ensure that the discussion is closely tied to the results. For example, I am not convinced the the results reported here show that "the children had developed a habit of using AST" (line 380) as a consequence of the intervention, which is implied.
Thank you for making a remark about this and thereby enabling us to specify this in the results. We have clarified that it is the participants that states that AST had become a habit as a consequence of the intervention, something that was evident in the interviews. Thereby we would like to keep the discussion about developing a habit of using AST.
Line 437 states that "Overall, these results underscore the importance of motivational considerations when explaining AST behavior". Can you explain more clearly what this means, and how you reach this conclusion?
We have added text about this in the discussion part.
The study limitations section should consider the implications of the fact that the school was attended by children of high SES.
Thank you for this remark, we have added text about this limitation in the discussion part.
Reviewer 3 Report
In this paper the authors provide further interesting information about the value of an active school transportation intervention using a qualitative research method.
In focus groups pupils, teachers and the principal informed how they perceived the three interventions (2016, 2017, and 2018) in order to promote active school transportation.
The introduction is well done. However, there is a discrepancy. In the introduction the focus is on the Social Cognitive Theory (SCT) described as the theoretical framework of the intervention. In the discussion, in contrast, all results are related to the Social Determination Theory (SDT). There should be an explanation why the theoretical framework has been changed.
Method:
2.1. It is not fully clear how and why the school and the class has been selected in 2016. Please briefly describe the selection procedure of the intervention group.
2.2. How is the gender distribution within the 40 pupils. Please include this information.
Results: At the beginning of the result section is some repetition. The subthemes are mentioned three times: In Table 1. in the last paragraph of page 4 and as titles on page 5, 6, and 8. Maybe the content can be organised differently.
The discussion is well done. Could be a limitation that the parents were not invited for a focus group?
Author Response
We appreciate the opportunity to improve our manuscript and we are grateful for your time and effort to make all of these remarks helping us to do that. Your comments are in bold and our answers are in italic.
Sincerely,
Eva Savolainen, Stina Rutberg, Ylva Backman and Anna-Karin Lindqvist.
In this paper the authors provide further interesting information about the value of an active school transportation intervention using a qualitative research method. In focus groups pupils, teachers and the principal informed how they perceived the three interventions (2016, 2017, and 2018) in order to promote active school transportation.
The introduction is well done. However, there is a discrepancy. In the introduction the focus is on the Social Cognitive Theory (SCT) described as the theoretical framework of the intervention. In the discussion, in contrast, all results are related to the Social Determination Theory (SDT). There should be an explanation why the theoretical framework has been changed.
Thank you for making us re-think this and we have added text in the discussion about the SCT since the result could indeed also be viewed through the SCT even though we believe that part of the result could be explained better with the SDT.
Method:
2.1. It is not fully clear how and why the school and the class has been selected in 2016. Please briefly describe the selection procedure of the intervention group.
Thank you for this remark, we have added text in the method section about this.
2.2. How is the gender distribution within the 40 pupils. Please include this information.
We have added information about this in the section Study design and Participants.
Results: At the beginning of the result section is some repetition. The subthemes are menttimes: In Table 1. in the last paragraph of page 4 and as titles on page 5, 6, and 8. Maybe the content can be organised differently.
We have removed the repetition of the subthemes on page 4.
The discussion is well done. Could be a limitation that the parents were not invited for a focus group?
We have added several limitations to the sections strength and limitations and asked for more research regarding parent’s choices regarding letting their children use or not use AST.